# Protocol for the evaluation of cost-effectiveness and health equity impact of a school-based tobacco prevention programme in a cluster randomised controlled trial (the TOPAS study)

Anni-Maria Pulkki-Brännström ,[1] Maria R Galanti ,[2] Maria Nilsson[1]

[1]Department of Epidemiology and Global Health, Umea University, Umea, Sweden
[2]Department of Global Public Health, Karolinska Institutet, Stockholm, Sweden

**Correspondence to**
Dr Anni-Maria Pulkki-Brännström;
anni-maria.pulkki-brannstrom@umu.se

## ABSTRACT

**Introduction** Despite a long-term downward trend in smoking prevalence, tobacco remains the number one risk factor for death and disability in Sweden. Globally, tobacco use generates a substantial economic burden for health systems and is also a major driver of socioeconomic inequalities in health. This article describes the planned cost-effectiveness and health equity impact evaluation of a multicomponent school-based programme to prevent the onset of tobacco use in adolescents.

**Methods and analysis** Cost-effectiveness of the multicomponent Tobacco-Free Duo programme will be evaluated against the educational component of the same programme only. An incremental cost-effectiveness ratio (ICER) will be calculated in terms of the cost per case prevented using the trial primary outcome and within-trial payer costs. If the ICER is negative, an incremental net benefit ratio will be calculated. Robustness of the results will be assessed through one-way sensitivity analyses. The slope index of inequality will be computed to assess the potential impact of the Tobacco-free Duo programme on education-related inequalities in the onset of smoking and in adult smoking cessation, comparing the two trial arms.

**Ethics and dissemination** Ethical approval was obtained from the Regional Ethics Review Board, Umeå (registration number 2017/255-31). The Public Health Agency of Sweden commissioned the study. The findings will be disseminated internationally within academia and to national and local policy-makers.

**Trial registration number** ISRCTN52858080; Pre-results.

## BACKGROUND

Tobacco use is a persistent global public health challenge, and scientific evidence of the effects of tobacco use on mortality and morbidity across the life course is robust. Around 50% of adult smokers die prematurely from tobacco-related disease.[1] Smoking is the single most preventable cause of death and one of the leading causes of inequity in health.[2] Furthermore, smoking poses large economic costs to individuals, healthcare systems and the society at large. The WHO

### Strengths and limitations of this study

► This study uses primary data and an experimental design to address the lack of rigorous economic evaluations of tobacco prevention programmes.
► The results will mirror the real costs and cost-effectiveness of the programme because the intervention is implemented by regular school personnel and integrated into schools' routine tobacco prevention activities.
► The study includes a health equity impact analysis in which the full programme (Tobacco-free Duo) is compared with structured classroom education only (E).
► Intervention effects are measured on the individual level, but resource use is measured on the level of the school.
► The study will not be able to capture the support that the adult 'pair' provides to the adolescent in keeping up the 'tobacco-free agreement'.

has estimated that the economic harm caused by smoking to the world economy is over US$500 billion each year.[3] The benefits from reduced tobacco use are undisputable and encompass improved population health, reduced health inequalities, reduced healthcare costs and improved economic productivity.

The young person becomes a smoker in a social context.[4] Many programmes to prevent young people's tobacco use employ methods based on social influence models. The essence of such models is to change the attitudes, knowledge and behaviour of young people in the context of a social environment. Parents, siblings and friends exert social influence, and the more smokers a young person has in the close context, the more he or she will start smoking.[5] Individual and contextual factors influence the process from initiation to regular smoking, and the

BMJ

individual risk of becoming a smoker is impacted by how the different factors relate to and interact with each other.

The need for evidence-based tobacco prevention programmes targeting young people is large. Programmes implemented outside the healthcare sector, such as in schools, potentially reach all adolescents and align well with global calls for multisectoral approaches to promoting health and well-being. However, relatively few rigorous evaluations exist. The Swedish agency responsible for recommendations in this area reviewed the literature and found the evidence base to be insufficient and many evaluation methodologies to be inappropriate, to support recommendations for interventions targeting young people.[6] A recent systematic review specifically of the health economic evidence on school-based interventions by Leão *et al*[7] found that few evaluations had been published, particularly not recently. There is a particular lack of economic evidence from studies that use recently collected primary data.[7]

The large socioeconomic inequalities in smoking are well documented, and parents' low socioeconomic status is a risk factor for adolescent smoking initiation.[8] However, a number of studies highlight that socioeconomic inequalities in tobacco use among adolescents are smaller than among adults.[9] Furthermore, some studies have found that where prevalence has decreased, socioeconomic inequalities have increased.[10] To our knowledge, no evaluation has as yet explicitly evaluated whether a prevention programme has an effect on socioeconomic inequalities in tobacco use among adolescents.

This protocol article describes the cost-effectiveness and health equity impact evaluation within the TOPAS (TObaksPreventivt Arbete i Skolan (in Swedish)) study. TOPAS is a 3-year intervention study that evaluates Tobacco-free Duo (T-Duo), a multicomponent school-based programme with public commitment to prevent the onset of tobacco use in adolescents in Sweden. The TOPAS study employs a mixed design approach and consists of a two-arm cluster randomised controlled trial (cRCT) and an observational study. Details of the TOPAS study and the mixed design effectiveness evaluation are described in Galanti *et al.*[11]

The planned study addresses several of the gaps identified in the evidence base. It is innovative in that an experimental design will be used to evaluate a universal preventive intervention from the perspectives of average health, cost-effectiveness, budget impact and impact on health inequalities.[12] It will provide data to predict system-level intervention costs.[7] This study will be the first to assess the cost-effectiveness of an intervention to prevent adolescent smoking initiation that includes (1) public commitment and (2) parents or other significant adults (other than school staff). Few previous studies have had a well-defined comparator; in our study, the comparator is similar to the one in Tengs *et al.*[13]

## AIM AND OBJECTIVES

The TOPAS study's health economic evaluation aims to measure the cost-effectiveness and health equity impact of the multicomponent T-Duo. The specific objectives are as follows:
1. To conduct a within-trial analysis of the incremental cost-effectiveness of T-Duo against the structured classroom education component of the same programme (E).
2. To assess whether socioeconomic inequalities in adolescent smoking initiation and adult smoking cessation differ between T-Duo and E.

## METHODS
### Study setting and population
The setting for the TOPAS study is secondary schools in nine regions in central Sweden, ranging from small rural communities to major cities. In Sweden, schools are funded through municipality taxes and run by either municipalities or private providers. Healthcare is funded and organised on a regional level. However, municipalities have some public health responsibilities in the area of tobacco prevention, including oversight of national laws and regulations around tobacco, and health promotion in schools.

The main study population consists of adolescents who were in seventh grade in the school year 2018/2019 and for whom active consent for data collection has been obtained from a guardian. Secondary study populations are smoking guardians of these adolescents and all adolescents in ninth grade in the three consecutive school years 2018/2019, 2019/2020 and 2020/2021.

### Trial design
In spring 2018, all lower secondary schools with at least two classes in the seventh to ninth grades, and located in 11 regions of central and south Sweden, were invited to participate in the study. An exclusion criterion was whether the school was already planning to adopt the T-Duo programme in the near future. The geographical area is large and includes urban, semi-urban and rural sparsely populated areas. School characteristics were collected as part of the baseline data collection to facilitate a comparison of how the schools that self-selected into the study compare with other schools nationally.

Thirty-four schools (clusters) were recruited into the trial and randomised to either the comprehensive T-Duo or structured classroom education only (E). The implementation period runs for the three school years from August 2018 to June 2021, when all data collection will be completed. All adolescents who started seventh grade in autumn 2018 are exposed to the intervention for 3 years; school staff are asked not to expose other year groups to the intervention during the study period. Detailed information about the school recruitment and randomisation processes, and a full description of the interventions are presented elsewhere.[11] A brief description of

the six components of the T-Duo programme, including the structured classroom education component that is common to both trial arms, and the support that the study team provides to schools for implementation are presented in the following sections.

## The multicomponent intervention with public commitment

In total, 34 schools were randomised to either the T-Duo programme (17 schools) or E (17 schools). Briefly, the overarching aim of the T-Duo programme is to prevent tobacco initiation among adolescents using schools as the main arena. T-Duo consists of six components. The first component is the tobacco-free pair, which is a signed agreement between the adolescent and a significant adult. Both parties promise to abstain from tobacco products at least until the adolescent finishes compulsory school. The adolescent is free to choose which significant adult to sign a contract with. Furthermore, the adult is free to choose how best to support the adolescent to keep up the agreement. The programme is introduced by the school staff to students (component 2) and to guardians at a regular parents' meeting (component 3). Adolescents who form a tobacco-free pair and sign a contract receive a membership card (component 4). The card is school-specific, and each school decides what benefits or discounts cardholders are entitled to; examples include t-shirts and discounts in the school cafeteria. At the end of each school year, all tobacco-free pairs are invited to disclose their tobacco-free status, and adolescents who affirm they have remained tobacco-free enter a prize draw organised by the school (component 5).

Structured classroom education is the sixth component of the T-Duo programme, and it is common to both trial arms (T-Duo and E). All schools receive lesson materials for two classroom sessions each term. These consist of age-tailored information and practical exercises about tobacco-related topics, for example, environmental impact. The materials have been externally commissioned for the study.

## Support to school staff for implementation

School staff are responsible for the implementation of the interventions in their school. School contact persons receive instructions and materials for each programme component from the study team. However, staff retain a degree of independence regarding the timing and exact delivery of the intervention components. Support for implementation is given once a year in the form of a training and networking event. The event is organised by the study team, but the training is delivered by external, professional trainers with specific expertise in tobacco prevention in schools. Information about topical tobacco-related issues is given, and upcoming lesson materials for the structured classroom education are introduced. The study team also informs about upcoming data collection. To facilitate data collection and retention of schools in the study, the study team is in regular contact with schools through monthly newsletters, email and telephone.

## Measurement of health outcomes

Health outcome data are collected from adolescents through questionnaires filled in at school and from guardians through postal questionnaires. Adolescents who change schools are followed up. Adolescents with missing data on the primary outcome will be excluded from the analysis.

The primary outcome in the cRCT is whether the adolescent has *never smoked cigarettes* (negative answer to the question: Did you ever try smoking, even a few puffs?) at the end of ninth grade (smoking onset). The primary outcome, which is self-reported, is measured in the main study population among adolescents whose guardians actively consented to data collection at baseline. Adolescents who report at baseline that they have tried smoking will be excluded from the analysis of the primary outcome. They are, however, exposed to and may participate in the intervention on equal terms with others who have never tried smoking.

The cost-effectiveness analysis will use the intention-to-treat results for the trial primary outcome, converted to the number of cases prevented. The average effect of T-Duo compared with E will be expressed as the change in the number of adolescents who have *ever smoked* by the end of ninth grade.

The equity impact analysis will use the trial primary outcome and one of the secondary outcomes, namely, parent has quit smoking (no smoking in the past 30 days). This outcome is also self-reported and measured among guardians who were smoking at baseline.

## Cost identification, measurement and valuation

Within-trial costs are collected prospectively from a payer perspective throughout the start-up and implementation phases of the cRCT. These costs consist of resource use (1) at the school level and (2) at the level of the programme.

School-level costs are collected primarily through an online data collection tool. A link to the survey is sent by email to the school's contact person approximately once a month during term time throughout the trial start-up and implementation periods. The survey contains questions about completed activities, staff time use and other direct costs for planning and delivery of the intervention. Questions are included to capture joint costs, such as time used for meetings. Where applicable, data reported by schools are complemented by records held by the study team. Examples of these data include attendance lists for the annual training and networking sessions, and the value of lottery prizes for T-Duo component 5. To value staff time use, average national salaries for the reported staff category will be used.

Programme costs are collected primarily from study financial accounts. These costs include development of intervention materials and annual training and networking sessions (eg, staff, refreshments, venue hire, travel). To not underestimate total intervention cost, a small proportion of the time used by the principal investigator and the research assistants who have direct contact

with schools is allocated to the intervention. This proportion will be subject to sensitivity analysis. Programme costs will thus cover direct costs and an allocated proportion of study joint costs.

All costs will be valued in Swedish kronor and inflated to 2021 values using the consumer price index. An annual 3% discount rate will be applied.

### Cost-effectiveness and budget impact analyses

Within-trial school-level and programme-level start-up and implementation costs for T-Duo and E will be added up, respectively. Total costs will be used to estimate unit costs per school, class and pupil. A within-trial incremental cost-effectiveness ratio will be calculated by dividing the incremental cost with the number of cases prevented, comparing T-Duo with E. If the resulting ratio is negative, incremental net benefit ratios will be calculated instead.[14]

The possibility of modelling budget impact and affordability of national scale-up of T-Duo will be explored. For this purpose, the existence of economies of scale at the level of the school will be analysed through the correlation between observed unit costs and the number of pupils or classes in each school.

To systematically explore uncertainty around cost-effectiveness ratios, one-way sensitivity analyses on key assumptions and programme-level parameters will be conducted. Thus, robustness of the within-trial results will be analysed with respect to, for example, staff salary levels, the allocation of joint costs and the inclusion of start-up costs.

Cost per case prevented will be compared with previously published estimates of preventive interventions for adolescents.[7] Up-to-date estimates of the societal costs of smoking will be provided to put the results in context. Cost-effectiveness will not be compared with threshold values, which are available in Sweden in terms of cost per quality-adjusted life-year (QALY) gained. We will thus not be able to rank T-Duo on the basis of cost-effectiveness against other interventions beyond prevention. We consider that presenting cost per case prevented, together with budget impact and other study outcomes, will provide sufficient evidence to inform scale-up by national and local decision-makers.

### Equity impact analysis

The slope index of inequality will be used to analyse whether the magnitude of socioeconomic inequalities in health outcomes differs between T-Duo and E. The highest education level attained by the adolescent's guardian(s) will be used to categorise adolescents and guardians from the lowest to the highest socioeconomic group. The slope index of inequality is a summary measure of absolute inequality that takes into account the relative size of each socioeconomic group.[15] The slope index will be adjusted for sex and computed for each health outcome (onset of smoking, adult smoking cessation) for T-Duo and E separately. If there is a statistically significant difference in the

index value between T-Duo and E, the hypothesis that T-Duo has no impact on health equity will be rejected.

### Patient and public involvement

The intervention under study is a preventive intervention for adolescents and is delivered by regular school staff. School staff are involved in data collection. No patients or members of the public were involved in designing the study.

## LIMITATIONS

The TOPAS study will provide rich individual-level and school-level data that can be used for further analyses. Of particular relevance may be analyses of costs and consequences beyond the study period, and analyses of relative health inequalities. No decision-analytic modelling is planned as part of this study. However, results for all individual-level health effects (whether positive or negative) among both adolescents and parents from the intention-to-treat analyses in the cRCT will be reported so that others may use these results for modelling and subsequently for comparing cost-effectiveness with a wider set of interventions. For example, if the trial finds a significant reduction in the onset of smoking, this result may be used as input into a model of the long-term cost-effectiveness of universal preventive interventions for adolescents, with health outcomes expressed as QALYs. Several models of this type were identified in the review by Leao *et al*.[7]

Common to many other cRCTs of complex public health interventions, individual-level sensitivity analysis will be neither possible nor conceptually desirable.[16] Intervention effects are measured on the individual level, but the intervention is delivered on the level of the school. Thus, exposure and resource use are also measured on the school level. The only exception is that exposure to the core component, the tobacco-free pair, is measured on the individual level.[11] However, the resource use associated with this component—what the adult pair does to support the adolescent in keeping up the agreement—will not be captured in this study.

### Ethics and dissemination

Ethical approval was obtained from the Regional Ethics Review Board, Umeå, prior to commencement of the study (registration number 2017/255-31). The results of the study will be disseminated in a scientific article in an international peer-reviewed journal. In addition, results will be disseminated to schools and decision-makers within Sweden through active participation in tobacco prevention-themed events.

**Acknowledgements** We want to thank the reviewer whose comments helped to clarify the purpose and scope of this protocol paper.

**Contributors** MRG and MN conceived of and designed the TOPAS study. A-MP-B and MN conceived the present study. MN acquired the funding. A-MP-B designed the study, wrote the first draft and coordinated the write-up of the paper. MN contributed substantially to writing the paper, and together with MRG revised the

work critically for important intellectual content. All authors approved of the final version and agree to be accountable for all aspects of the work.

**Funding** The study was funded by the Public Health Agency of Sweden, with case reference number 01346-2017 2.3.2.

**Competing interests** None declared.

**Patient and public involvement** Patients and/or the public were not involved in the design, or conduct, or reporting, or dissemination plans of this research.

**Patient consent for publication** Not required.

**Provenance and peer review** Not commissioned; externally peer reviewed.

**ORCID iDs**
Anni-Maria Pulkki-Brännström http://orcid.org/0000-0001-8723-8131
Maria R Galanti http://orcid.org/0000-0002-7805-280X

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
