## [Reviewer comments · BMJ Open]

ARTICLE DETAILS

TITLE (PROVISIONAL)	Protocol for the evaluation of the cost-effectiveness and health equity impact of a school-based tobacco prevention programme in a cluster randomised controlled trial (The TOPAS Study)
AUTHORS	Pulkki-Brännström, Anni-Maria; Galanti, Maria; Nilsson, Maria

VERSION 1 – REVIEW

REVIEWER	Michael Maciosek HealthPartners Institute
REVIEW RETURNED	24-Jan-2021

GENERAL COMMENTS	Some clarifications and additional details are needed prior to publication. If the authors' purpose in publishing this protocol is to increase confidence that analyses were conducted as planned and conclusions were not tailored retrospectively to match the results, then the missing pieces are substantial omissions. The paper could be shortened to make room for the missing details. Some examples of shortening are provided here. A careful review and edit by the authors is likely to identify others. I believe some of the examples here would both shorten the paper and improve reader comprehension. 1) After reading several pages, the relationship between the Tobacco-Free Duo programme (TFD) and the TOPAs seems straight forward, but I was confused until I got several pages in. It would seem simple enough to not make the reader struggle with this for that long. The problem starts with the abstract methods section. It mentions the TOPAS study in the first sentence and in the next sentence jumps to mentioning the TFD, without explanation of the relationship between the two. This leaves the reader confused about what the protocol is for. There seems to be no reason, from the reader's perspective, to mention TOPAS in the abstract if the protocol is about TFD. Thus, the first sentence of the abstract methods section could simply be deleted, leaving the introduction of TOPAS to the body of the paper. Similarly, the fourth paragraph of the background section would be more clear if it started by directly stating that the protocol covers the CE and equity evaluation of the Tobacco-Free Duo programme, which is part of the larger TOPAS study. The last sentence of the fourth paragraph could be moved to be the first sentence with a small modification for this purpose. Alternatively, it might be more clear (and I think correct) to say that this protocol describes the CE and health equity evaluation of TOPAS, and then describe TFD as the intervention component of TOPAS. 2) Somewhat related, the paper can be shortened by describing TOPAS and TFD just once. The information that is now the fourth paragraph of Background could be moved to methods section where TOPAS and TFD are also described, and the combined text could be shortened.
--

3) The last paragraph of 'Aim and objectives' is really background. It could be moved to background with the other text there that motivates the analysis, with tightening of the motivation.

4) The subheadings for the brief sections under Methods seem unnecessary, and in some cases interrupt flow causing unnecessary words. For example, the second of two paragraphs under 'Trial Design' seems more like an introduction to the next section ('The multicomponent...') and less about design. The second, long, sentence of that paragraph could be deleted with the next paragraph (with no header) starting simply, 'Briefly, the overarching aim...'

5) There are other places to tighten. I would like to point the authors to the long discussion section in particular. This may be personal bias, but I do not understand what value is provided by a discussion section for a protocol other than highlighting foreseeable limitations. Even that is more useful when published with the results. Therefore, I would recommend considerable cuts and tightening of the Discussion to make room for the missing information noted next. Additional information / clarification needed.

6) A direct statement about the start date, or a study timeline figure would be helpful. Currently the second paragraph of the study population section notes that those who were in 7th grade in 2018/19 is the main study population. In the next section, text states "The intervention started with the training school staff in August 2018. The implementation period runs until 2021,..." The reader is left to piece together information from two different sections and infer that implementation started at the beginning of the school year when students arrived and ran for 3 years. Why not just say so, all in the same place?.

7) How were the 34 schools selected and recruited? Was the geographic area limited? Do you believe them to be representative of the geographic area from which they were selected, or might self-selection make them less representative than if a random subset of schools were selected?

8) How is tobacco use defined? The text seems to use tobacco use and smoking interchangeably in some parts. The measurement section states that the primary outcome is never smoked. Never smoked what? Any combustible tobacco or cigarettes only? What about vaping? When any tobacco is the measure, what does that include – any nicotine product?

9) What qualifies as never smoking? Not even a puff, ever?

10) How will smoking/tobacco use be ascertained. Self-report? If self-report, will there be any verification, like cotinine levels or reports from individuals who know the study subject? If there is no verification and rewards for being tobacco free, does not that create a strong incentive for deception in the main outcome in the TFD schools that could bias results?

11) When assessing whether or not a study ever smoked by the end of 9th grade, will the investigators assume that all adolescents started the 7th grade as never smokers?

12) What criteria (threshold) will be used to determine whether the program was cost-effective. Without pre-specifying this, the authors could write favorable conclusions for a wide range of results. If the authors can't pre-specify a threshold and are only providing the information for decision-makers to interpret, with the authors commit to not characterizing the results in publications?

13) What is the statistical analysis plan for assessing the equity outcome? How does the statistical analysis plan address clustering? Will any statistically significant difference between groups in the noted equity outcomes be reported as inequities? If so, what is the anticipated statistical power to detect differences between groups?

	14) The cost-effectiveness plan for long-term cost effectiveness appears to fall short of the usual approach of evaluating CE in a computational decision-analysis model. It is not clear how published estimates of LYs and costs associated with tobacco use can be used as described. Examples of the studies the authors envision using and how the long-term cost and LYs will be computed for the two study arms using those studies would be helpful. It is not evident how one can go from 9th grade smoking status to lifetime outcomes with any accuracy without modeling smoking initiation from 9th grade through young adult years and cessation, and relapse after that, along with competing causes of death. Other questions for authors: 15) Why were the details of the CE study and equity impact not included in the original trial protocol (Galanti et al.)? So much space seems to be spent giving study background relative to the additional information about these other outcomes. It seems it would have been better to provide one complete protocol. 16) Was this study overseen by an institutional / human subjects review board? If yes, which? If not, why? Perhaps this information was provided in separate fields in the online submission and would be provided to readers with publication. It was not provided to reviewers.
--	--

VERSION 1 – AUTHOR RESPONSE

REVIEWER: 1

Some clarifications and additional details are needed prior to publication. If the authors' purpose in publishing this protocol is to increase confidence that analyses were conducted as planned and conclusions were not tailored retrospectively to match the results, then the missing pieces are substantial omissions. The paper could be shortened to make room for the missing details. Some examples of shortening are provided here. A careful review and edit by the authors is likely to identify others. I believe some of the examples here would both shorten the paper and improve reader comprehension.

Response (0): Our study will produce rich individual- and school-level data that will likely give rise to several new research questions and post hoc analyses once the main results become known. We consider that following revisions made in response to the reviewer's comments, our significantly shortened protocol now better signals which analyses were planned a priori. Another purpose with this paper is to describe the overall objectives and specific components of the evaluation that we think are of more general interest for other health economists in designing a cluster RCT evaluation of a preventive public health intervention. This latter point is elaborated on below under Response 15, regarding the value of a separate health economic protocol paper.

1) After reading several pages, the relationship between the Tobacco-Free Duo programme (TFD) and the TOPAs seems straight forward, but I was confused until I got several pages in. It would seem simple enough to not make the reader struggle with this for that long. The problem starts with the abstract methods section. It mentions the TOPAS study in the first sentence and in the next sentence jumps to mentioning the TFD, without explanation of the relationship between the two. This leaves the reader confused about what the protocol is for. There seems to be no reason, from the reader's perspective, to mention TOPAS in the abstract if the protocol is about TFD. Thus, the first sentence of the abstract methods section could simply be deleted, leaving the introduction of TOPAS to the body of the paper. Similarly, the fourth paragraph of the background section would be more clear if it started by directly stating that the protocol covers the CE and equity evaluation of the Tobacco-Free Duo programme, which is part of the larger TOPAS study. The last sentence of the fourth paragraph could be moved to be the first sentence with a small modification for this purpose. Alternatively, it

might be more clear (and I think correct) to say that this protocol describes the CE and health equity evaluation of TOPAS, and then describe TFD as the intervention component of TOPAS.

Response (1): We thank the reviewer for these concrete suggestions. We have deleted the first sentence of the Abstract and subsequently the TOPAS study is now not mentioned in the Abstract. We revised the fourth paragraph of the Background section, which now begins as follows:

This protocol paper describes the cost-effectiveness and health equity impact evaluation within the TOPAS study. TOPAS is a three-year intervention study that evaluates Tobacco-free Duo, a multi-component school-based programme with public commitment to prevent the onset of tobacco use in adolescents in Sweden. The TOPAS study employs a mixed design approach and consists of a two-arm cluster randomised controlled trial (cRCT) and an observational study.

2) Somewhat related, the paper can be shortened by describing TOPAS and TFD just once. The information that is now the fourth paragraph of Background could be moved to methods section where TOPAS and TFD are also described, and the combined text could be shortened.

Response (2): The fourth paragraph of the Background has been deleted except for one sentence, which is now in revised form in the Methods section on page 7:

All adolescents who started 7th grade in autumn 2018 are exposed to the intervention for three years; school staff are specifically asked not to expose other year groups to the intervention during the study period.

3) The last paragraph of 'Aim and objectives' is really background. It could be moved to background with the other text there that motivates the analysis, with tightening of the motivation.

Response (3): The paragraph has been moved up as suggested. The second half of the Background has been revised to better motivate the analysis. Some arguments have been moved here from the Discussion, which has been cut down. One new reference has been added on page 5 to better motivate the equity impact analysis, as follows:

Furthermore, some studies have found that where prevalence has decreased, socioeconomic inequalities have increased (e.g. (10)). To our knowledge, no evaluation has as yet explicitly evaluated whether a prevention programme has an effect on socioeconomic inequalities in tobacco use among adolescents.

4) The subheadings for the brief sections under Methods seem unnecessary, and in some cases interrupt flow causing unnecessary words. For example, the second of two paragraphs under 'Trial Design' seems more like an introduction to the next section ('The multicomponent...') and less about design. The second, long, sentence of that paragraph could be deleted with the next paragraph (with no header) starting simply, 'Briefly, the overarching aim....'

Response (4): Several sub-headings have been removed and sub-sections have been merged in the Methods section. In addition, the two specific changes suggested here have been made.

5) There are other places to tighten. I would like to point the authors to the long discussion section in particular. This may be personal bias, but I do not understand what value is provided by a discussion section for a protocol other than highlighting foreseeable limitations. Even that is more useful when published with the results. Therefore, I would recommend considerable cuts and tightening of the Discussion to make room for the missing information noted next.

Response (5): We note that BMJ Open Author Guidelines for protocol papers do not specify a Discussion section, however, it is included in several previously published papers that include a health economic component, e.g. Feldman et al 2017 <http://bmjopen.bmj.com/content/6/8/e011202>.

We have decided to keep a Discussion section, however it has been cut down to two paragraphs and renamed Limitations.

Additional information / clarification needed.

6) A direct statement about the start date, or a study timeline figure would be helpful. Currently the second paragraph of the study population section notes that those who were in 7th grade in 2018/19 is the main study population. In the next section, text states “The intervention started with the training school staff in August 2018. The implementation period runs until 2021,...” The reader is left to piece together information from two different sections and infer that implementation started at the beginning of the school year when students arrived and ran for 3 years. Why not just say so, all in the same place?.

Response (6): Several start-up activities were ongoing already in spring 2018, however implementation on the school level did not begin until the autumn. The time period is now more clearly described on page 7:

The implementation period runs for the three school years from August 2018 to June 2021, when all data collection will be completed.

7) How were the 34 schools selected and recruited? Was the geographic area limited? Do you believe them to be representative of the geographic area from which they were selected, or might self-selection make them less representative than if a random subset of schools were selected?

Response (7): The sample of schools enrolled in the study cannot be assumed to be a representative random sample of all eligible schools in the area, because of self-selection. The following description of the recruitment of schools has been added on pages 6-7:

In spring 2018, all lower secondary schools with at least two classes in the 7th to 9th grades, and located in 11 regions of central and south Sweden, were invited to participate in the study. An exclusion criteria was whether the school was already planning to adopt the Tobacco-free Duo Programme in the near future. The geographical area is large, includes both urban, semi-urban and rural, sparsely populated areas. School characteristics were collected as part of the baseline data collection to facilitate a comparison of how the schools who self-selected into the study compare with other schools nationally.

8) How is tobacco use defined? The text seems to use tobacco use and smoking interchangeably in some parts. The measurement section states that the primary outcome is never smoked. Never smoked what? Any combustible tobacco or cigarettes only? What about vaping? When any tobacco is the measure, what does that include – any nicotine product?

Response (8): The primary outcome, never smoked, is defined as that the adolescent reports never to have tried cigarettes, not even a puff. Secondary outcomes cover other types of tobacco use. With the exception of cessation among smoking guardians, the revised version of the manuscript no longer refers to other secondary outcome measures (see response 14 below). We have revised the terminology and now consistently refer to smoking initiation and smoking onset in the paper.

9) What qualifies as never smoking? Not even a puff, ever?

Response (9): Yes. This has been clarified on pages 8-9:

The primary outcome in the cRCT is whether the adolescent has never smoked cigarettes (negative answer to the question: Did you ever try smoking, even a few puffs?) at the end of 9th grade (smoking onset). The primary outcome, which is self-reported, is measured in the main study population among adolescents whose guardians actively consented to data collection at baseline.

10) How will smoking/tobacco use be ascertained. Self-report? If self-report, will there be any verification, like cotinine levels or reports from individuals who know the study subject? If there is no verification and rewards for being tobacco free, does not that create a strong incentive for deception in the main outcome in the TFD schools that could bias results?

Response (10): All outcomes in the study are self-reported, and we have clarified this on page 9: The primary outcome, which is self-reported, is measured in the main study population among adolescents whose guardians actively consented to data collection at baseline.

...

The equity impact analysis uses the trial primary outcome and one of the secondary outcomes, namely parent has quit smoking (no smoking in the past 30 days). This outcome is also self-reported and measured among guardians who were smoking at baseline.

We cannot exclude that self-report reliability may differ between intervention and control schools, and this will be discussed as a concern when reporting the results from the effectiveness evaluation. However, the observational study conducted alongside the RCT will collect data from reference schools that are not part of the RCT, and analyse the general development of progression of tobacco use from 7th to 9th grade. Thus the observational study results will facilitate a discussion of to what extent such bias is likely to be present in the intervention schools in the RCT study.

Two other aspects are worth noting. First, the financial value of the incentives provided to adolescents for being tobacco-free are modest, and limited to those who sign a contract. The primary outcome analysis uses instead an intention-to-treat approach, in which all adolescents including those who do not sign a contract, are included. Second, biochemical verification methods used for adult smokers are not of value in an adolescent population in which children who smoke, smoke rarely or very little.

11) When assessing whether or not a study ever smoked by the end of 9th grade, will the investigators assume that all adolescents started the 7th grade as never smokers?

Response (11): No. Rather, adolescents who at baseline report having already tried cigarette smoking will be excluded from the analysis. This has been added on page 9: Adolescents who report at baseline that they have tried smoking, will be excluded from the analysis of the primary outcome. They are however be exposed to and may participate in the intervention on equal terms with others who have never tried smoking.

12) What criteria (threshold) will be used to determine whether the program was cost-effective. Without pre-specifying this, the authors could write favorable conclusions for a wide range of results. If the authors can't pre-specify a threshold and are only providing the information for decision-makers to interpret, with the authors commit to not characterizing the results in publications?

Response (12): In background documents informing development of national guidelines, the Swedish National Board of Health and Welfare publicises thresholds it uses to categorise cost per QALY gained as low/moderate/high/very high (<https://www.socialstyrelsen.se/en/regulations-and-guidelines/national-guidelines/>). However, these are not applicable to our study since our cost-effectiveness analysis will be limited to the within-trial results and thus to cost per case prevented (see below Response 14). The following has been added on pages 10-11: Cost per case prevented will be compared with previously published estimates of preventive interventions for adolescents (7). Up-to-date estimates of the societal costs of smoking will be provided to put the results in context. Cost-effectiveness will not be compared with threshold values, which are available in Sweden in terms of cost per QALY gained. We will thus not be able to rank T-Duo on the basis of cost-effectiveness against other interventions beyond prevention. We consider

that presenting cost per case prevented, together with budget impact and the other study outcomes, will provide sufficient evidence to inform scale-up by national and local decision-makers.

13) What is the statistical analysis plan for assessing the equity outcome? How does the statistical analysis plan address clustering? Will any statistically significant difference between groups in the noted equity outcomes be reported as inequities? If so, what is the anticipated statistical power to detect differences between groups?

Response (13): The main methodological choices are the use of a summary measure of inequality and comparing intervention with control. In a recent evaluation of another universal prevention programme (Pulkki-Brännström et al. 2020, <https://jech.bmj.com/content/74/7/605>) the same overall approach was used to evaluate equity impact. The Equity impact analysis – subsection has been rewritten to provide more details about the analytical methods. It now reads as follows:

The slope index of inequality will be used to analyse whether the magnitude of socioeconomic inequalities in the health outcomes differs between T-Duo and E. The highest education level attained by the adolescent's guardian(s) will be used to categorise adolescents and guardians from lowest to highest socioeconomic group. The slope index of inequality is a summary measure of absolute inequality that takes into account the relative size of each socioeconomic group (14). The slope index will be adjusted for sex and computed for each health outcome (onset of smoking, adult smoking cessation) for T-Duo and E separately. If there is a statistically significant difference in the index value between T-Duo and E, the hypothesis that T-Duo has no impact on health equity will be rejected.

As reflected in the above text, the reviewer's questions regarding our plans to evaluate whether the intervention has any impact on health inequalities were particularly valuable. We have made some changes to the manuscript in response and these are detailed below.

First, regarding sample size, the RCT has been powered to detect a difference in the onset of smoking between intervention and control; we have not conducted a sample size calculation to assess whether it has enough power to detect differences in socioeconomic inequalities between the trial arms. However, it is worthwhile to note that statistical power is not limited by the categorisation of study participants into socioeconomic groups because our chosen analytical method, the slope index of inequality, uses the whole sample.

However, regarding the planned school-level analyses of inequality, it seems likely the sample size (number of schools in the observational study) is not big enough to answer the question whether the intervention has any impact on health inequalities. We have revised the manuscript so that only the two individual-level health outcomes from the RCT are used to study the equity impact of the intervention. Thus, we have removed references to the observational study and a comparison with usual practice since these data are now not used in any of the analyses described in the manuscript.

Second, we do not plan to adjust for clustering in the health equity impact analysis. We do not see conceptually what adjusting for clustering would imply when the analysis aims to quantify the extent of inequality using a summary measure. Furthermore, while the slope index of inequality is widely used in studies of health inequalities outside the health economics discipline, we have not been able to find methodological studies in which it is used on data from a cluster RCT, and are thus not aware of previous studies in which the index would have been adjusted to account for clustering.

14) The cost-effectiveness plan for long-term cost effectiveness appears to fall short of the usual approach of evaluating CE in a computational decision-analysis model. It is not clear how published estimates of LYs and costs associated with tobacco use can be used as described. Examples of the studies the authors envision using and how the long-term cost and LYs will be computed for the two study arms using those studies would be helpful. It is not evident how one can go from 9th grade

smoking status to lifetime outcomes with any accuracy without modeling smoking initiation from 9th grade through young adult years and cessation, and relapse after that, along with competing causes of death.

Response (14): As we have set out in the manuscript, the within-trial results are the primary analysis. We had set out a secondary analysis as a separate objective, in which existing literature – largely decision-analytic models – would be used to convert the cost per case prevented to cost per life-years saved. We chose this approach because constructing a decision-analytic model of our own is beyond the scope of our study. At the same time, we acknowledge that presenting results as cost per case prevented limits the extent to which our results can be compared with other types of interventions.

In response to the reviewer's concern about the feasibility of a long-term analysis without decision-analytic modelling, we have decided to remove the second objective from this study protocol. Thus the protocol is explicitly limited to the analyses that we in practice intend (and have the resources) to conduct, namely the within-trial cost-effectiveness and health equity impact analyses.

References to cost per life-year saved have been removed from the Abstract, which now reads as follows:

An incremental cost-effectiveness ratio (ICER) will be calculated in terms of the cost per case prevented using the trial primary outcome and within-trial payer costs. If the ICER is negative, an incremental net benefit ratio will be calculated.

The Aim and objectives read as follows:

The TOPAS study's health economic evaluation aims to measure the cost-effectiveness and health equity impact of the multi-component Tobacco-free Duo programme (T-Duo). The specific objectives are:

1. To conduct a within-trial analysis of the incremental cost-effectiveness of T-Duo against the structured classroom education component of the same programme (E).
2. To assess whether socioeconomic inequalities in adolescent smoking initiation and in adult smoking cessation differ between T-Duo, E and current practice.

In Methods, the paragraph describing analyses beyond the trial perspective have been removed.

In the Discussion, we have added a paragraph that acknowledges the limitation of within-trial analysis but also the potential use of our results as inputs into decision-analytic models:

The TOPAS study will provide rich individual- and school-level data that can be used for further analyses. Of particular relevance may be analyses of costs and consequences beyond the study period; and analyses of relative health inequalities. No decision-analytic modelling is planned as part of this study. However, results for all individual-level health effects (whether positive or negative) among both adolescents and parents from the intention-to-treat analyses in the cRCT will be reported so that others may use these results for modelling and subsequently for comparing cost-effectiveness with a wider set of interventions. For example, if the trial finds a significant reduction in the onset of smoking, this result may be used as input into a model of the long-term cost-effectiveness of universal preventive interventions for adolescents, with health outcomes expressed as quality-adjusted life-years. Several models of this type were identified in the review by Leao et al. (7).

Other questions for authors:

15) Why were the details of the CE study and equity impact not included in the original trial protocol (Galanti et al.)? So much space seems to be spent giving study background relative to the additional

information about these other outcomes. It seems it would have been better to provide one complete protocol.

Response (15): We believe that for cluster RCTs of complex public health interventions, a separate health economic protocol paper can contribute towards a discussion about the scope of health economics in such evaluations. We have been informed by the publication in BMJ Open of two health economic protocol papers of cluster RCTs of complex public health interventions within global health: Skordis-Worrall et al. (2016) <http://bmjopen.bmj.com/content/6/11/e012046> and Haghparast-Bidgoli et al. (2018) <https://bmjopen.bmj.com/content/8/8/e022035> .

In our paper, we cite Weatherly et al (2009), who argued that cost-effectiveness, budget impact, and impact on health inequalities, should all be considered by analysts studying public health interventions. In our view this is still not current practice for health economic studies, in particular RCTs, which are relatively rare for evaluating public health interventions in high income countries. Thus, the value of a separate health economic protocol paper is that it sets out the overall approach which can inform others on how a study can be planned to include all the components mentioned by Weatherly et al..

A separate protocol paper describing the health economic aspects of the study was planned from the very beginning of the project, and is briefly mentioned in both the Trial registration and in Galanti et al. (2020). The protocol paper by Galanti et al. (2020) describes the TOPAS study as a whole and provides an overview of the evaluation approach. A detailed description is provided of the Effectiveness evaluation, in which the experimental (RCT) design is combined with an observational design. Short paragraphs describe evaluation components related to Intervention implementation and fidelity; and Adolescents' experiences and reported side effects of the intervention. In contrast, we considered that more detail was warranted regarding the design of the health economic aspects of the study.

16) Was this study overseen by an institutional / human subjects review board? If yes, which? If not, why? Perhaps this information was provide in separate fields in the online submission and would be provided to readers with publication. It was not provided to reviewers.

Response (16): The study was approved by the Regional Ethics Review Board, Umeå. This was mentioned in the Abstract but not in the main manuscript. To clarify, and following an Editorial request, the final subsection of the manuscript is now titled "Ethics and dissemination" and includes the following statement:

Ethical approval was obtained from the Regional Ethics Review Board, Umeå, prior to commencement of the study (Registration number 2017/255-31).

VERSION 2 – REVIEW

REVIEWER	Michael Maciosek HealthPartners Institute
REVIEW RETURNED	16-Apr-2021
GENERAL COMMENTS	I appreciate the thorough response to reviewer comments and the changes to the manuscript. I have no additional substantial comments. Please note that I had intended my question about accounting for clustering in the analysis to be more focused on the measure of adolescent smoking initiation than on health equity. However, that is a measure of the main study. Accounting for clustering in main analysis is implied by the sample size discussion of the main protocol, and therefore need not be addressed in this

	manuscript. My apologies for any confusion caused by my unclear question.
--	---